# Neural Jump Stochastic Differential Equations

**Junteng Jia**
Cornell University
jj585@cornell.edu

**Austin R. Benson**
Cornell University
arb@cs.cornell.edu

## Abstract

Many time series are effectively generated by a combination of deterministic continuous flows along with discrete jumps sparked by stochastic events. However, we usually do not have the equation of motion describing the flows, or how they are affected by jumps. To this end, we introduce Neural Jump Stochastic Differential Equations that provide a data-driven approach to learn continuous and discrete dynamic behavior, i.e., hybrid systems that both flow and jump. Our approach extends the framework of Neural Ordinary Differential Equations with a stochastic process term that models discrete events. We then model temporal point processes with a piecewise-continuous latent trajectory, where the discontinuities are caused by stochastic events whose conditional intensity depends on the latent state. We demonstrate the predictive capabilities of our model on a range of synthetic and real-world marked point process datasets, including classical point processes (such as Hawkes processes), awards on Stack Overflow, medical records, and earthquake monitoring.

## 1 Introduction

In a wide variety of real-world problems, the system of interest evolves continuously over time, but may also be interrupted by stochastic events [1, 2, 3]. For instance, the reputation of a Stack Overflow user may gradually build up over time and determines how likely the user gets a certain badge, while the event of earning a badge may steer the user to participate in different future activities [4]. How can we simultaneously model these continuous and discrete dynamics?

One approach is with hybrid systems, which are dynamical systems characterized by piecewise continuous trajectories with a finite number of discontinuities introduced by discrete events [5]. Hybrid systems have long been used to describe physical scenarios [6], where the equation of motion is often given by an ordinary differential equation. A simple example is table tennis — the ball follows physical laws of motion and changes trajectory abruptly when bouncing off paddles. However, for problems arising in social and information sciences, we usually know little about the time evolution mechanism. And in general, we also have little insight about how the stochastic events are generated.

Here, we present Neural Jump Stochastic Differential Equations (JSDEs) for learning the continuous and discrete dynamics of a hybrid system in a data-driven manner. In particular, we use a latent vector $\mathbf{z}(t)$ to encode the state of a system. The latent vector $\mathbf{z}(t)$ flows continuously over time until an event happens at random, which introduces an abrupt jump and changes its trajectory. The continuous flow is described by Neural Ordinary Differential Equations (Neural ODEs), while the event conditional intensity and the influence of the jump are parameterized with neural networks as functions of $\mathbf{z}(t)$.

The Neural ODEs framework models continuous transformation of a latent vector as an ODE flow and parameterizes the flow dynamics with a neural network [7]. The approach is a continuous analogy to residual networks, ones with infinite depth and infinitesimal step size, which brings about many desirable properties. Remarkably, the derivative of the loss function can be computed via the adjoint method, which integrates the adjoint equation backwards in time with *constant memory* regardless of the network depth. However, the downside of these continuous models is that they cannot incorporate discrete events (or inputs) that abruptly change the latent vector. To address this limitation, we extend

the Neural ODEs framework with discontinuities for modeling hybrid systems. In particular, we show how the discontinuities caused by discrete events should be handled in the adjoint method. More specifically, at the time of a discontinuity, not only does the latent vector describing the state of the system changes abruptly; as a consequence, the adjoint vector representing the loss function derivatives also jumps. Furthermore, our Neural JSDE model can serve as a stochastic process for generating event sequences. The latent vector $\mathbf{z}(t)$ determines the conditional intensity of event arrival, which in turn causes a discontinuity in $\mathbf{z}(t)$ at the time an event happens.

A major advantage of Neural JSDEs is that they can be used to model a variety of marked point processes, where events can be accompanied with either a discrete value (say, a class label) or a vector of real-valued features (e.g., spatial locations); thus, our framework is broadly applicable for time series analysis. We test our Neural JSDE model in a variety of scenarios. First, we find that our model can learn the intensity function of a number of classical point processes, including Hawkes processes and self-correcting processes (which are already used broadly in modeling, e.g., social systems [8, 9, 10]). After, we show that Neural JSDEs can achieve state-of-the-art performance in predicting discrete-typed event labels, using datasets of awards on Stack Overflow and medical records. Finally, we demonstrate the capabilities of Neural JSDEs for modeling point processes where events have real-valued feature vectors, using both synthetic data as well as earthquake data, where the events are accompanied with spatial locations as features.

## 2 Background, Motivation, and Challenges

In this section, we review classical temporal point process models and the Neural ODE framework of Chen et al. [7]. Compared to a discrete time step model like an RNN, the continuous time formation of Neural ODEs makes it more suitable for describing events with real-valued timestamps. However, Neural ODEs enforce continuous dynamics and therefore cannot model sudden event effects.

### 2.1 Classical Temporal Point Process Models

A temporal point process is a stochastic generative model whose output is a sequence of discrete events $\mathcal{H} = \{\tau_j\}$. An event sequence can be formally described by a counting function $N(t)$ recording the number of events before time $t$, which is defined as follows:

$$N(t) = \sum_{\tau_j \in \mathcal{H}} H(t - \tau_j), \quad \text{where} \quad H(t) = \begin{cases} 0 & t \leq 0 \\ 1 & \text{otherwise,} \end{cases} \tag{1}$$

where $H$ is the Heaviside step function. Oftentimes, we are interested in a temporal point process whose future outcome depends on historical events [11]. Such dependency is best described by a conditional intensity function $\lambda(t)$. Let $\mathcal{H}_t$ denote the subset of events up to but not including $t$. Then $\lambda(t)$ defines the probability density of observing an event conditioned on the event history:

$$\mathbb{P}\{\text{event in } [t, t + dt) \mid \mathcal{H}_t\} = \lambda(t) \cdot dt \tag{2}$$

Using this form, we now describe some of the most well-studied point process models, which we later use in our experiments.

**Poisson processes.** The conditional intensity is a function $g(t)$ independent of event history $\mathcal{H}_t$. The simplest case is a homogeneous Poisson process where the intensity function is a constant $\lambda_0$:

$$\lambda(t) = g(t) = \lambda_0. \tag{3}$$

**Hawkes processes.** These processes assume that events are self-exciting. In other words, an event leads to an increase in the conditional intensity function, whose effect decays over time:

$$\lambda(t) = \lambda_0 + \alpha \sum_{\tau_j \in \mathcal{H}_t} \kappa(t - \tau_j), \tag{4}$$

where $\lambda_0$ is the baseline intensity, $\alpha > 0$, and $\kappa$ is a kernel function. We consider two widely used kernels: (1) the exponential kernel $\kappa_1$, which is often used for its computational efficiency [12]; and (2) the power-law kernel $\kappa_2$, which is used for modeling in seismology [13] and social media [14]:

$$\kappa_1(t) = e^{-\beta t}, \qquad \kappa_2(t) = \begin{cases} 0 & t < \sigma \\ \frac{\beta}{\sigma} \left(\frac{t}{\sigma}\right)^{-\beta-1} & \text{otherwise.} \end{cases} \tag{5}$$

The variant of the power-law kernel we use here has a delaying effect.

**Self-correcting processes.** A self-correcting process assumes the conditional intensity grows exponentially with time and an event suppresses future events. This model has been used for modeling earthquakes once aftershocks have been removed [15]:

$$\lambda(t) = e^{\mu t - \beta N(t)}. \tag{6}$$

**Marked temporal point processes.** Oftentimes, we care not only about when an event happens, but also *what* the event is; having such labels makes the point process *marked*. In these cases, we use a vector embedding $\mathbf{k}$ to denote event type, and $\mathcal{H} = \{(\tau_j, \mathbf{k}_j)\}$ for an event sequence, where each tuple denotes an event with embedding $\mathbf{k}_j$ happening at timestamp $\tau_j$. This setup is applicable to events with discrete types as well as events with real-valued features. For discrete-typed events, we use a one-hot encoding $\mathbf{k}_j \in \{0, 1\}^m$, where $m$ is the number of discrete event types. Otherwise, the $\mathbf{k}_j$ are real-valued feature vectors.

## 2.2 Neural ODEs

A Neural ODE defines a continuous-time transformation of variables [7]. Starting from an initial state $\mathbf{z}(t_0)$, the transformed state at any time $t_i$ is given by integrating an ODE forward in time:

$$\frac{d\mathbf{z}(t)}{dt} = f(\mathbf{z}(t), t;\ \theta), \qquad \mathbf{z}(t_i) = \mathbf{z}(t_0) + \int_{t_0}^{t_i} \frac{d\mathbf{z}(t)}{dt} dt. \tag{7}$$

Here, $f$ is a neural network parameterized by $\theta$ that defines the ODE dynamics.

Assuming the loss function depends directly on the latent variable values at a sequence of checkpoints $\{t_i\}_{i=0}^N$ (i.e., $\mathcal{L} = \mathcal{L}(\{\mathbf{z}(t_i)\};\ \theta)$), Chen et al. proposed to use the adjoint method to compute the derivatives of the loss function with respect to the initial state $\mathbf{z}(t_0)$, model parameters $\theta$, and the initial time $t_0$ as follows. First, define the initial condition of the adjoint variables as follows,

$$\mathbf{a}(t_N) = \frac{\partial \mathcal{L}}{\partial \mathbf{z}(t_N)}, \qquad \mathbf{a}_\theta(t_N) = 0, \qquad \mathbf{a}_t(t_N) = \frac{\partial \mathcal{L}}{\partial t_N} = \mathbf{a}(t_N) f(\mathbf{z}(t_N), t_N;\ \theta). \tag{8}$$

Then, the loss function derivatives $d\mathcal{L}/d\mathbf{z}(t_0) = \mathbf{a}(t_0)$, $d\mathcal{L}/d\theta = \mathbf{a}_\theta(t_0)$, and $d\mathcal{L}/dt_0 = \mathbf{a}_t(t_0)$ can be computed by integrating the following ordinary differential equation backward in time:

$$\frac{d\mathbf{a}(t)}{dt} = -\mathbf{a}(t) \frac{\partial f(\mathbf{z}(t), t;\ \theta)}{\partial \mathbf{z}(t)}, \quad \mathbf{a}(t_0) = \mathbf{a}(t_N) + \int_{t_N}^{t_0} \left[ \frac{d\mathbf{a}(t)}{dt} - \sum_{i \neq N} \delta(t - t_i) \frac{\partial \mathcal{L}}{\partial \mathbf{z}(t_i)} \right] dt$$

$$\frac{d\mathbf{a}_\theta(t)}{dt} = -\mathbf{a}(t) \frac{\partial f(\mathbf{z}(t), t;\ \theta)}{\partial \theta}, \quad \mathbf{a}_\theta(t_0) = \mathbf{a}_\theta(t_N) + \int_{t_N}^{t_0} \frac{d\mathbf{a}_\theta(t)}{dt} dt$$

$$\frac{d\mathbf{a}_t(t)}{dt} = -\mathbf{a}(t) \frac{\partial f(\mathbf{z}(t), t;\ \theta)}{\partial t}, \quad \mathbf{a}_t(t_0) = \mathbf{a}_t(t_N) + \int_{t_N}^{t_0} \left[ \frac{d\mathbf{a}_t(t)}{dt} - \sum_{i \neq N} \delta(t - t_i) \frac{\partial \mathcal{L}}{\partial t_i} \right] dt. \tag{9}$$

Although solving Eq. (9) requires the value of $\mathbf{z}(t)$ along its entire trajectory [7], $\mathbf{z}(t)$ can be recomputed backwards in time together with the adjoint variables starting with its final value $\mathbf{z}(t_N)$ and therefore induce no memory overhead.

## 2.3 When can Neural ODEs Model Temporal Point Processes?

The continuous Neural ODE formulation makes it a good candidate for modeling events with real-valued timestamps. In fact, Chen et al. applied their model for learning the intensity of Poisson processes, which notably do not depend on event history. However, in many real-world applications, the event (e.g., financial transactions or tweets) often provides feedback to the system and influences the future dynamics [16, 17].

There are two possible ways to encode the event history and model event effects. The first approach is to parametrize $f$ with an explicit dependence on time: events that happen before time $t$ changes the function $f$ and consequently influence the trajectory $\mathbf{z}(t)$ after time $t$. Unfortunately, even the mild

assumption requiring $f$ to be finite would imply the event effects "kick in" continuously, and therefore cannot model events that create immediate shocks to a system (e.g., effects of Federal Reserve interest rate changes on the stock market). For this reason, areas such as financial mathematics have long advocated for discontinuous time series models [18, 19]. The second alternative is to encode the event effects as abrupt jumps of the latent vector $\mathbf{z}(t)$. However, the original Neural ODE framework assumes a Lipschitz continuous trajectory, and therefore cannot model temporal point processes that depend on event history (such as a Hawkes process).

In the next section, we show how to incorporate jumps into the Neural ODE framework for modeling event effects, while maintaining the simplicity of the adjoint method for training.

## 3  Neural Jump Stochastic Differential Equations

In our setup, we are given a sequence of events $\mathcal{H} = \{(\tau_j, \mathbf{k}_j)\}$ (i.e., a set of tuples, each consisting of a timestamp and a vector), and we are interested in both simulating and predicting the likelihood of future events.

### 3.1  Latent Dynamics and Stochastic Events

At a high level, our model represents the latent state of the system with a vector $\mathbf{z}(t) \in \mathbb{R}^n$. The latent state continuously evolves with a deterministic trajectory until interrupted by a stochastic event. Within any time interval $[t, t + dt)$, an event happens with the following probability:

$$\mathbb{P}\{\text{event happens in } [t, t + dt) \mid \mathcal{H}_t\} = \lambda(t) \cdot dt, \tag{10}$$

where $\lambda(t) = \lambda(\mathbf{z}(t))$ is the total conditional intensity for events of all types. The embedding of an event happening at time $t$ is sampled from $\mathbf{k}(t) \sim p(\mathbf{k}|\mathbf{z}(t))$. Here, both $\lambda(\mathbf{z}(t))$ and $p(\mathbf{k}|\mathbf{z}(t))$ are parameterized with neural networks and learned from data. In cases where events have discrete types, $p(\mathbf{k}|\mathbf{z}(t))$ is supported on the finite set of one-hot encodings and the neural network directly outputs the intensity for every event. On the other hand, for events with real-valued features, we parameterize $p(\mathbf{k}|\mathbf{z}(t))$ with a Gaussian mixture model, whose parameters $\eta$ depend on $\mathbf{z}(t)$. The mapping from $\mathbf{z}(t)$ to $\eta$ is learned with another neural network.

Next, let $N(t)$ be the number of events up to time $t$. The latent state dynamics of our Neural JSDE model is described by the following equation:

$$d\mathbf{z}(t) = f(\mathbf{z}(t), t;\ \theta) \cdot dt + w(\mathbf{z}(t), \mathbf{k}(t), t;\ \theta) \cdot dN(t), \tag{11}$$

where $f$ and $w$ are two neural networks that control the flow and jump, respectively. Following our definition for the counting function (Eq. (1)), all time dependent variables are left continuous in $t$, i.e., $\lim_{\epsilon \to 0+} \mathbf{z}(t - \epsilon) = \mathbf{z}(t)$. Section 3.3 describes the neural network architectures for $f$, $w$, $\lambda$, and $p$.

Now that we have fully defined the latent dynamics and stochastic event handling, we can simulate the hybrid system by integrating Eq. (11) forward in time with an adaptive step size ODE solver. The complete algorithm for simulating the hybrid system with stochastic events is described in Appendix A.1. However, in this paper, we focus on prediction instead of simulation.

### 3.2  Learning the Hybrid System

For a given set of model parameters, we compute the log probability density for a sequence of events $\mathcal{H} = \{(\tau_j, \mathbf{k}_j)\}$ and define the loss function as

$$\mathcal{L} = -\log \mathbb{P}(\mathcal{H}) = -\sum_j \log \lambda(\mathbf{z}(\tau_j)) - \sum_j \log p(\mathbf{k}_j|\mathbf{z}(\tau_j)) + \int_{t_0}^{t_N} \lambda(\mathbf{z}(t))dt. \tag{12}$$

In practice, the integral in Eq. (12) is computed by a weighted sum of intensities $\lambda(\mathbf{z}(t_i))$ on checkpoints $\{t_i\}$. Therefore, computing the loss function $\mathcal{L} = \mathcal{L}(\{\mathbf{z}(t_i)\};\ \theta)$ requires integrating Eq. (11) forward from $t_0$ to $t_N$ and recording the latent vectors along the trajectory.

The loss function derivatives are evaluated with the adjoint method (Eq. (9)). However, we encounter jumps in the latent vector $\Delta \mathbf{z}(\tau_i) = w(\mathbf{z}(\tau_j), \mathbf{k}_j, \tau_j;\ \theta)$ when integrating the adjoint equations backwards in time (Fig. 1). These jumps also introduce discontinuities to the adjoint vectors at $\tau_j$.

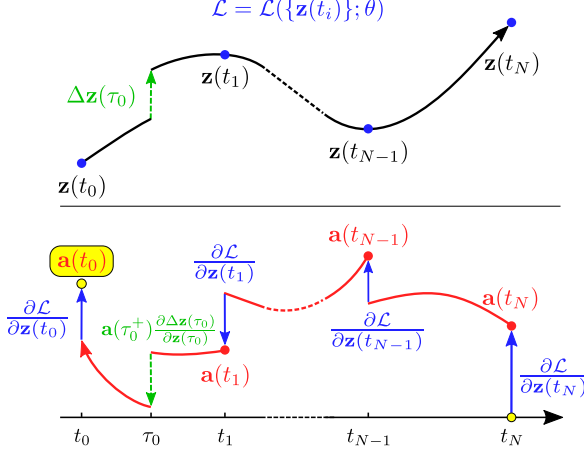

Figure 1: Reverse-mode differentiation of an ODE with discontinuities. Each jump $\Delta \mathbf{z}(\tau_j)$ in the latent vector (green, top panel) also introduces a discontinuity for adjoint vectors (green, bottom panel).

Denote the right limit of any time dependent variable $x(t)$ by $x(t^+) = \lim_{\epsilon \to 0^+} x(t + \epsilon)$. Then, at any timestamp $\tau_j$ when an event happens, the left and right limits of the adjoint vectors $\mathbf{a}$, $\mathbf{a}_\theta$, and $\mathbf{a}_t$ exhibit the following relationships (see Appendix A.2 for the derivation):

$$\mathbf{a}(\tau_j) = \mathbf{a}(\tau_j^+) + \mathbf{a}(\tau_j^+) \frac{\partial \left[ w(\mathbf{z}(\tau_j), \mathbf{k}_j, \tau_j; \theta) \right]}{\partial \mathbf{z}(\tau_j)}$$

$$\mathbf{a}_\theta(\tau_j) = \mathbf{a}_\theta(\tau_j^+) + \mathbf{a}(\tau_j^+) \frac{\partial \left[ w(\mathbf{z}(\tau_j), \mathbf{k}_j, \tau_j; \theta) \right]}{\partial \theta}$$

$$\mathbf{a}_t(\tau_j) = \mathbf{a}_t(\tau_j^+) + \mathbf{a}(\tau_j^+) \frac{\partial \left[ w(\mathbf{z}(\tau_j), \mathbf{k}_j, \tau_j; \theta) \right]}{\partial \tau_j}. \tag{13}$$

In order to compute the loss function derivatives $d\mathcal{L}/d\mathbf{z}(t_0) = \mathbf{a}(t_0)$, $d\mathcal{L}/d\theta = \mathbf{a}_\theta(t_0)$, and $d\mathcal{L}/dt_0 = \mathbf{a}_t(t_0)$, we integrate the adjoint vectors backwards in time following Eq. (9). However, at every $\tau_j$ when an event happens, the adjoint vectors is discontinuous and needs to be lifted from its right limit to its left limit. One caveat is that computing the Jacobian in Eq. (13) requires the value of $\mathbf{z}(\tau_j)$ at the left limit, which need to be recorded during forward integration. The complete algorithm for integrating $\mathbf{z}(t)$ forward and $\mathbf{z}(t), \mathbf{a}(t), \mathbf{a}_\theta(t), \mathbf{a}_t(t)$ backward is described in Appendix A.3.

### 3.3 Network Architectures

Figure 2 shows the network architectures that parameterizes our model. In order to better simulate the time series, the latent state $\mathbf{z}(t) \in \mathbb{R}^n$ is further split into two vectors: $\mathbf{c}(t) \in \mathbb{R}^{n_1}$ encodes the internal state, and $\mathbf{h}(t) \in \mathbb{R}^{n_2}$ encodes the memory of events up to time $t$, where $n = n_1 + n_2$.

**Dynamics function** $f(\mathbf{z})$**.** We parameterize the internal state dynamics $\partial \mathbf{c}(t)/\partial t$ by a multi-layer perceptron (MLP). Furthermore, we require $\partial \mathbf{c}(t)/\partial t$ to be orthogonal to $\mathbf{c}(t)$. This constrains the internal state dynamics to a sphere and improves the stability of the ODE solution. On the other hand, the event memory $\mathbf{h}(t)$ decays over time, with a decay rate parameterized by another MLP, whose output passes through a softplus activation to guarantee the decay rate to be positive.

**Jump function** $w(\mathbf{z}(t))$**.** An event introduces a jump $\Delta \mathbf{h}(t)$ to event history $\mathbf{h}(t)$. The jump is parameterized by a MLP that takes the event embedding $\mathbf{k}(t)$ and internal state $\mathbf{c}(t)$ as input. Our architecture also assumes that the event does not directly interrupt internal state (i.e., $\Delta \mathbf{c}(t) = 0$).

**Intensity** $\lambda(\mathbf{z}(t))$ **and probability** $p(\mathbf{k}|\mathbf{z}(t))$**.** We use a MLP to compute both the total intensity $\lambda(\mathbf{z}(t))$ and the probability distribution over the event embedding. For events that are discrete (where $\mathbf{k}$ is a one-hot encoding), the MLP directly outputs the intensity of each event type. For events with real-valued features, the probability density distribution is represented by a mixture of Gaussians, and the MLP outputs the weight, mean, and variance of each Gaussian.

## 4 Experimental Results

Next, we use our model to study a variety of synthetic and real-world time series of events that occur at real-valued timestamps. We train all of our models on a workstation with a 8 core i7-

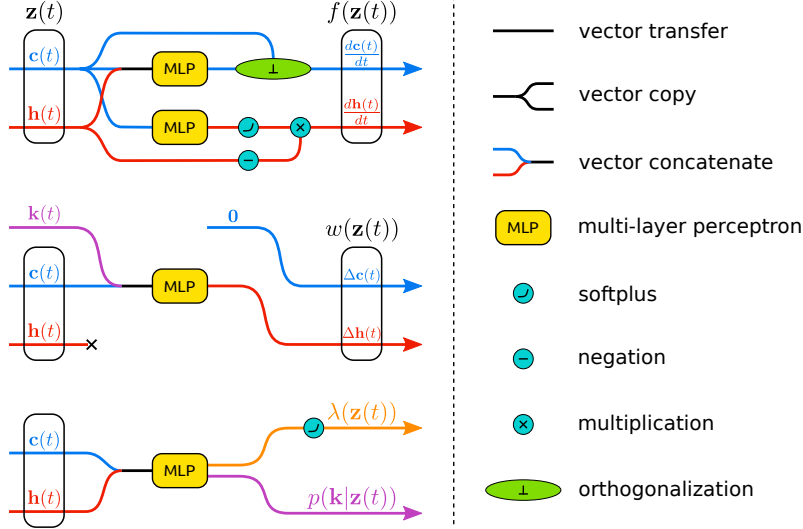

Figure 2: Neural network architectures that map latent vector $\mathbf{z}(t)$ to $f(\mathbf{z}(t))$, $w(\mathbf{z}(t))$, $\lambda(\mathbf{z}(t))$, and $p(\mathbf{k}|\mathbf{z}(t))$. The vectors and computations flow from left to right. The round-cornered rectangles (yellow) is a fully connected multi-layer perceptron with CELU activation function. The circles (cyan) are element-wise operations such as negation, multiplication, and softplus activation. The ellipse (green) represents the projection that takes the output of the multi-layer perceptron and orthogonalizes it against $\mathbf{c}$. The colors of the curves encode the dimensionality of the vectors.

7700 CPU @ 3.60GHz processor and 32 GB memory. We use the Adam optimizer with $\beta_1 = 0.9, \beta_2 = 0.999$; the architectures, hyperparameters, and learning rates for different experiments are reported below. The complete implementation of our algorithms and experiments are available at https://github.com/000Justin000/torchdiffeq/tree/jj585.

## 4.1 Modeling Conditional Intensity — Synthetic Data from Classical Point Process Models

We first demonstrate our model's flexibility to capture the influence of event history on the conditional intensity in a variety of point processes models. To show the robustness of our model, we consider the following generative processes (we only focus on modeling the conditional intensity in this part, so all events are assigned the same type): (i) Poisson Process: the conditional intensity is given by $\lambda(t) = \lambda_0$, where $\lambda_0 = 1.0$; (ii) Hawkes Process (Exponential Kernel): the conditional intensity is given by Eq. (4) with the exponential kernel $\kappa_1$ in Eq. (5), where $\lambda_0 = 0.2$, $\alpha = 0.8$, $\beta = 1.0$; (iii) Hawkes Process (Power-Law Kernel): the conditional intensity is given by Eq. (4) with the power-law kernel $\kappa_2$ in Eq. (5), where $\lambda_0 = 0.2$, $\alpha = 0.8$, $\beta = 2.0$, $\sigma = 1.0$; and (iv) Self-Correcting Process: the conditional intensity is given by Eq. (6), where $\mu = 0.5$, $\beta = 0.2$.

For each generative process, we create a dataset by simulating $500$ event sequences within the time interval $[0, 100]$ and use $60\%$ for training, $20\%$ for validation and $20\%$ for testing. We fit our Neural JSDE model to each dataset using the training procedure described above, using a $5$-dimensional latent state ($n_1 = 3$, $n_2 = 2$) and MLPs with one hidden layer of $20$ units (the learning rate for the Adam optimizer is set to be $10^{-3}$ with weighted decay rate $10^{-5}$). In addition, we fit the parameters of each of the four point processes to each dataset using maximum likelihood estimation as implemented in PtPack.[1] These serve as baselines for our model. Furthermore, we also compare the performance of our model with an RNN. The RNN models the intensity function on 2000 evenly spaced timestamps across the entire time window (using a $20$-dimensional latent state and tanh activation), and each event time is rounded to the closest timestamp.

The average conditional intensity varies among different generative models. For a meaningful comparison, we measure accuracy with the mean absolute percentage error:

$$\frac{1}{t_1 - t_0} \int_{t_0}^{t_1} dt \, |\frac{\lambda^*_{\text{model}}(t) - \lambda^*_{\text{GT}}(t)}{\lambda^*_{\text{GT}}(t)}| \times 100\%, \tag{14}$$

Table 1: The mean absolute percentage error of the predicted conditional intensities. Each column represents a different type of generating process. Each row represents a prediction model. In all cases, our neural JSDE outperforms the RNN baseline by a significant margin.

|  | Poisson | Hawkes (E) | Hawkes (PL) | Self-Correcting |
|---|---|---|---|---|
| Poisson | 0.1 | 188.2 | 95.6 | 29.1 |
| Hawkes (E) | 0.3 | 3.5 | 155.4 | 29.1 |
| Hawkes (PL) | 0.1 | 128.5 | 9.8 | 29.1 |
| Self-Correcting | 98.7 | 101.0 | 87.1 | 1.6 |
| RNN | 3.2 | 22.0 | 20.1 | 24.3 |
| Neural JSDE | 1.3 | 5.9 | 17.1 | 9.3 |

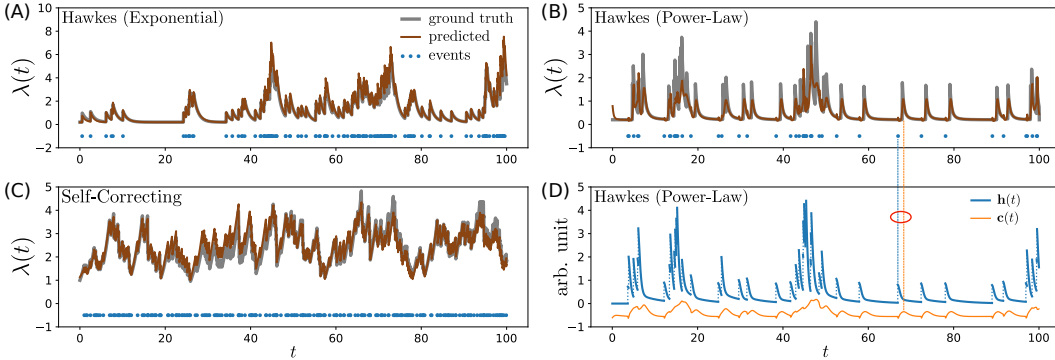

Figure 3: The ground truth and predicted conditional intensity of three event sequences generated by different processes (A–C), and an example of the latent state dynamics (D). Each blue dot represents an event at the corresponding time. In all cases, our model captures the general trends in the intensity.

where $\lambda_{\text{model}}^*(t)$ is the trained model intensity and $\lambda_{\text{GT}}^*(t)$ is the ground truth intensity. The integral is approximated by evaluation at 2000 uniformly spaced points (the same points that are used for training the RNN model). Table 1 reports the errors of the conditional intensity for our model and the baselines. In all cases, our neural JSDE model is a better fit for the data than the RNN and other point process models (except for the ground truth model, which shows what we can expect to achieve with perfect knowledge of the process). Figure 3 shows how the learned conditional intensity of our Neural JSDE model effectively tracks the ground truth intensities. Remarkably, our model is able to capture the delaying effect in the power-law kernel (Fig. 3D) through a complex interplay between the internal state and event memory: although an event immediately introduces a jump to the event memory $\mathbf{h}(t)$, the intensity function peaks when the internal state $\mathbf{c}(t)$ is the largest, which lags behind $\mathbf{h}(t)$.

## 4.2 Discrete Event Type Prediction on social and medical datasets.

Next, we evaluate our model on a discrete-type event prediction task with two real-world datasets. The Stack Overflow dataset contains the awards history of 6633 users in an online question-answering website [21]. Each sequence is a collection of badges a user received over a period of 2 years, and there are 22 different badges types in total. The medical records (MIMIC2) dataset contains the clinical visit

Table 2: The classification error rate of our model on discrete event type prediction. The baseline error rates are taken form [20].

| Error Rate | [21] | [20] | NJSDE |
|---|---|---|---|
| Stack Overflow | 54.1 | 53.7 | **52.7** |
| MIMIC2 | 18.8 | **16.8** | 19.8 |

history of 650 de-identified patients in an Intensive Care Unit [21]. Each sequence consists of visit events of a patient over 7 years, where event type is the reason for the visit (75 reasons in total). Using 5-fold cross validation, we predict the event type of every held-out event $(\tau_j, \mathbf{k}_j)$ by choosing the event embedding with the largest probability $p(\mathbf{k}|\mathbf{z}(\tau_j))$ given the past event history $\mathcal{H}_{\tau_j}$. For the Stack Overflow dataset, we use a 20-dimensional latent state ($n_1 = 10$, $n_2 = 10$) and MLPs with one hidden layer of 32 units to parameterize the dynamics function $f$, $w$, $\lambda$, $p$. For MIMIC2, we use

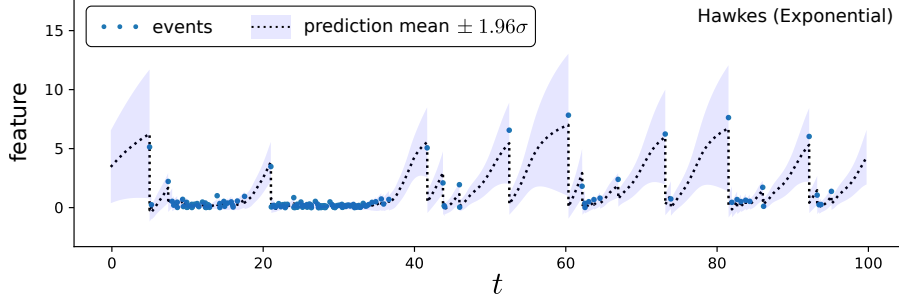

Figure 4: The ground truth and predicted event embedding in one event sequences.

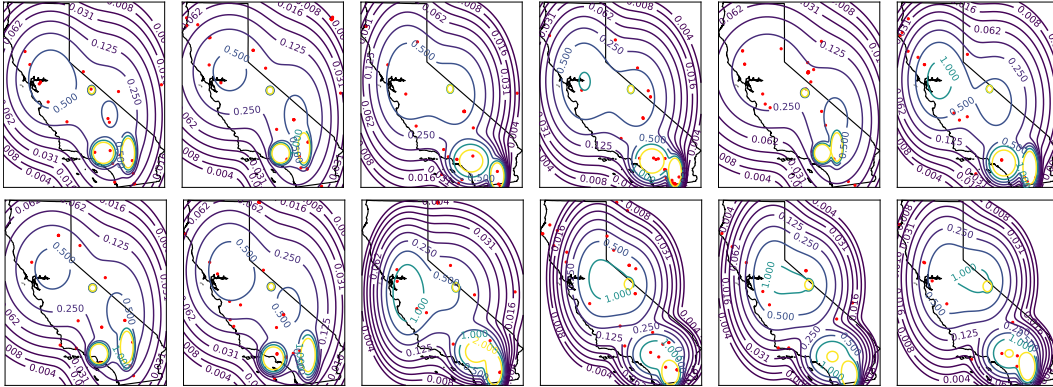

Figure 5: Contour plots of the predicted conditional intensity and the locations of earthquakes (red dots) over the years 2007-2018 using earthquake training data from 1970–2006.

a $64$-dimensional latent state ($n_1 = 32$, $n_2 = 32$) and MLPs with one hidden layer of $64$ units. The learning rate is set to be $10^{-3}$ with weighted decay rate $10^{-5}$.

We compare the event type classification accuracy of our model against two other models for learning event sequences that directly simulate the next event based on the history, namely neural point process models based on an RNN [21] or LSTM [20]. Note that these approaches model event sequences but not trajectories. Our model not only achieve similar performance with these state-of-the-art models for discrete event type prediction (Table 2), but also allows us to model events with real-valued features, as we study next.

### 4.3 Real-Valued Event Feature Prediction — Synthetic and Earthquake data

Next, we use our model to predict events with real-valued features. To this end, we first test our model on synthetic event sequences whose event times are generated by a Hawkes process with exponential kernel, but the feature of each event records the time interval since the previous event. We train our model in a similar way as to Section 4.1, using a 10-dimensional latent state ($n_1 = 5$, $n_2 = 5$) and MLPs with one hidden layer of 20 units (the learning rate for Adam optimizer is set to be $10^{-4}$ with weighted decay rate $10^{-5}$). This achieves a mean absolute error of $0.353$. In contrast, the baseline of simply predicting the mean of the features seen so far has an error of $3.654$. Figure 4 shows one event sequence and predicted event features.

Finally, we provide an illustrative example of real-world data with real-valued features. We use our model to predict the time and locations of earthquakes above level 4.0 in 2007–2018 using historical data from 1970–2006.[2] In this case, an event's features are the longitude and latitude locations of an earthquake. This time, we use a 20-dimensional latent state ($n_1 = 10$, $n_2 = 10$), and parameterize the event feature's probability density distribution by a mixture of 5 Gaussians. Figure 5 shows the contours of the conditional intensity of the learned Neural JSDE model.

## 5 Related Work

**Modeling point processes.** Temporal point processes are elegant abstractions for time series analysis. The self-exciting nature of the Hawkes process has made it a key model within machine learning and information science [22, 23, 24, 25, 26, 27, 28, 29]. However, classical point process models (including the Hawkes process) make strong assumptions about how the event history influences future dynamics. To get around this, RNNs and LSTMs have been adapted to directly model events as time steps within the model [21, 20]. However, these models do not consider latent space dynamics in the absence of events as we have, which may reflect time-varying internal evolution that inherently exists in the system. Xiao et al. also proposed a combined approach to event history and internal state evolution by simultaneously using two RNNs — one that takes event sequence as input, and one that models evenly spaced time intervals [30]. In contrast, our model provides a unified approach that addresses both aspects by making a connection to ordinary differential equations and can be efficiently trained with the adjoint method using only constant memory. Another approach uses GANs to circumvent modeling the intensity function [31]; however, this cannot provide insight into the dynamics in the system. Most related to our approach is the recently proposed ODE-RNN model [32], which uses an RNN to make abrupt updates to a hidden state that mostly evolves continuous.

**Learning differential equations.** More broadly, learning parameters of differential equations from data has been successful for physics-based problems with deterministic dynamics [33, 34, 35, 36, 37]. In terms of modeling real-world randomness, Wang et al. introduced a jump-diffusion stochastic differential equation framework for modeling user activities [38], parameterizing the conditional intensity and opinion dynamics with a fixed functional form. More recently, Ryder et al. proposed an RNN-based variational method for learning stochastic dynamics [39], which has later been generalized for infinitesimal step size [40, 41]. These approaches focus on randomness introduced by Brownian motion, which cannot model abrupt jumps of latent states. Finally, recent developments of robust software packages [42, 43] and numerical methods [44] have made the process of learning model parameters easier and more reliable for a host of models.

## 6 Discussion

We have developed Neural Jump Stochastic Differential Equations, a general framework for modeling temporal event sequences. Our model learns both the latent continuous dynamics of the system and the abrupt effects of events from data. The model maintains the simplicity and memory efficiency of Neural ODEs and uses a similar adjoint method for learning; in our case, we additionally model jumps in the trajectory with a neural network, and handle the effects of this discontinuity in the learning method. Our approach is quite flexible, being able to model intensity functions and discrete or continuous event types, all while providing interpretable latent space dynamics.

**Acknowledgements.** This research was supported by NSF Award DMS-1830274, ARO Award W911NF19-1-0057, and ARO MURI.

## Footnotes

[1] https://github.com/dunan/MultiVariatePointProcess

[2]Data from https://www.kaggle.com/danielpe/earthquakes

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
