[Supplementary Material · Neural_Jump_Stochastic_Differential_Equation_SI.pdf]

# A Appendix

## A.1 Algorithm for Simulating Hybrid System with Stochastic Events

---
**Algorithm 1:** Dynamics simulation for hybrid system

---
**Input** : model parameter $\theta$, start time $t_0$, end time $t_N$, initial state $\mathbf{z}(t_0)$
**Output :** event sequence $\mathcal{H}$

initialize $t = t_0, \ j = 0, \ \mathcal{H} = \{\}, \ \mathbf{z} = \mathbf{z}(t_0)$
**while** $t < t_N$ **do**
    $dt = \text{AdpativeForwardStepSize}(\mathbf{z}, t, \theta)$           ▷ from ODE solver
    $(\tau_j, \mathbf{k}_j) = \text{SimulateNextEvent}(\mathbf{z}, t, \theta)$      ▷ sample exponential distribution
    **if** $\tau_j > t + dt$ **then**
        $\mathbf{z} = \text{StepForward}(\mathbf{z}, dt, \theta)$           ▷ $1^{\text{st}}$ term in Eq. (11)
    **else**
        $\mathcal{H} = \mathcal{H} \cup \{(\tau_j, \mathbf{k}_j)\}$           ▷ record event
        $j = j + 1$
        $dt = \tau_j - t$           ▷ shrink step size
        $\mathbf{z} = \text{StepForward}(\mathbf{z}, dt, \theta)$
        $\mathbf{z} = \text{JumpForward}(\mathbf{z}, (\tau_j, \mathbf{k}_j), \theta)$      ▷ $2^{\text{nd}}$ term in Eq. (11)
    **end**
    $t = t + dt$
**end**

---

Note that when an event $i$ happens within the step size $dt$ proposed by the ODE solver, $dt$ needs to shrink so that $t + dt$ is no larger than $\tau_i$.

## A.2 Adjoint Sensitivity Analysis at Discontinuities

When the $j^{\text{th}}$ event happens at timestamp $\tau_j$, the left and right limits of latent variables are related by,

$$\mathbf{z}(\tau_j^+) = \mathbf{z}(\tau_j) + w(\mathbf{z}(\tau_j), \mathbf{k}_j, \tau_j; \ \theta) \tag{15}$$

where all the time dependent variables are left continuous in time. According to Remark 2 from [45], the left and right limits of adjoint sensitivity variables at a discontinuity satisfy

$$\mathbf{a}(\tau_j) = \mathbf{a}(\tau_j^+) \left( \frac{\partial \mathbf{z}(\tau_j^+)}{\partial \mathbf{z}(\tau_j)} \right). \tag{16}$$

Substituting Eq. (15) in Eq. (16) gives,

$$\begin{aligned} \mathbf{a}(\tau_j) &= \mathbf{a}(\tau_j^+) \left( \mathbf{I} + \frac{\partial w(\mathbf{z}(\tau_j), \mathbf{k}_j, \tau_j; \ \theta)}{\partial \mathbf{z}(\tau_j)} \right) \\ &= \mathbf{a}(\tau_j^+) + \mathbf{a}(\tau_j^+) \frac{\partial \left[ w(\mathbf{z}(\tau_j), \mathbf{k}_j, \tau_j; \ \theta) \right]}{\partial \mathbf{z}(\tau_j)}. \end{aligned} \tag{17}$$

Moreover, Eq. (16) can be generalized to obtain the jump of $\mathbf{a}_\theta$ and $\mathbf{a}_t$ at the discontinuities. In the work of Chen et al. [7], the authors define an augmented latent variables and its dynamics as,

$$\mathbf{z}_{\text{aug}}(t) = \begin{bmatrix} \mathbf{z} \\ \theta \\ t \end{bmatrix}(t), \quad \frac{d\mathbf{z}_{\text{aug}}(t)}{dt} = f_{\text{aug}}(\mathbf{z}, t; \ \theta) = \begin{bmatrix} f(\mathbf{z}, t; \ \theta) \\ \mathbf{0} \\ 1 \end{bmatrix}, \quad \mathbf{a}_{\text{aug}}(t) = \begin{bmatrix} \mathbf{a} & \mathbf{a}_\theta & \mathbf{a}_t \end{bmatrix}(t). \tag{18}$$

Following the same convention, we define the augmented jump function at $\tau_j$ as,

$$w_{\text{aug}}(\mathbf{z}(\tau_j), \mathbf{k}_j, \tau_j; \ \theta) = \begin{bmatrix} w(\mathbf{z}(\tau_j), \mathbf{k}_j, \tau_j; \ \theta) \\ \mathbf{0} \\ 0 \end{bmatrix}. \tag{19}$$

We can verify that the left and right limits of the augmented latent variables satisfy

$$\mathbf{z}_{\mathrm{aug}}(\tau_j^+) = \begin{bmatrix} \mathbf{z}(\tau_j) \\ \theta \\ \tau_j \end{bmatrix} + \begin{bmatrix} w(\mathbf{z}(\tau_j), \mathbf{k}_j, \tau_j;\ \theta) \\ \mathbf{0} \\ 0 \end{bmatrix} = \mathbf{z}_{\mathrm{aug}}(\tau_j) + w_{\mathrm{aug}}(\mathbf{z}(\tau_j), \mathbf{k}_j, \tau_j;\ \theta). \qquad (20)$$

The augmented dynamics is only a special case of the general Neural ODE framework, and the jump of adjoint variables can be calculated as

$$\mathbf{a}_{\mathrm{aug}}(\tau_j) = \mathbf{a}_{\mathrm{aug}}(\tau_j^+) \left( \frac{\partial \mathbf{z}_{\mathrm{aug}}(\tau_j^+)}{\partial \mathbf{z}_{\mathrm{aug}}(\tau_j)} \right) = [\mathbf{a} \quad \mathbf{a}_\theta \quad \mathbf{a}_t] (\tau_j^+) \begin{bmatrix} \mathbf{I} + \frac{\partial w}{\partial \mathbf{z}(\tau_j)} & \frac{\partial w}{\partial \theta} & \frac{\partial w}{\partial \tau_j} \\ \mathbf{0} & \mathbf{I} & \mathbf{0} \\ 0 & 0 & 1 \end{bmatrix}, \qquad (21)$$

which is equivalent to Eq. (13).

## A.3 Algorithm for Adjoint Method with Discontinuities

---

**Algorithm 2:** Algorithm for computing the loss function and its derivatives

---

**Input** : model parameter $\theta$, start time $t_0$, end time $t_N$, initial state $\mathbf{z}(t_0)$, event sequence $\mathcal{H}$
**Output :** loss function $\mathcal{L}$ and derivatives $d\mathcal{L}/d\mathbf{z}(t_0) = \mathbf{a}(t_0)$, $d\mathcal{L}/d\theta = \mathbf{a}_\theta(t_0)$,
        $d\mathcal{L}/dt_0 = \mathbf{a}_t(t_0)$

initialize $t = t_0$, $\mathbf{z} = \mathbf{z}(t_0)$
**while** $t < t_N$ **do**
     $dt = \mathtt{AdpativeForwardStepSize}(\mathbf{z}, t, \theta)$                    ▷ from ODE solver
     $(\tau_j, \mathbf{k}_j) = \mathtt{GetNextEvent}(\mathcal{H}, t)$             ▷ find next event in sequence
     **if** $\tau_j > t + dt$ **then**
         $\mathbf{z} = \mathtt{StepForward}(\mathbf{z}, dt, \theta)$                ▷ $1^{\mathrm{st}}$ term in Eq. (11)
     **else**
         $dt = \tau_j - t$                         ▷ shrink step size
         $\mathbf{z} = \mathtt{StepForward}(\mathbf{z}, dt, \theta)$
         $\mathbf{z} = \mathtt{JumpForward}(\mathbf{z}, (\tau_j, \mathbf{k}_j), \theta)$         ▷ $2^{\mathrm{nd}}$ term in Eq. (11)
     **end**
     $t = t + dt$
**end**

$\mathcal{L} = \mathcal{L}\left(\{\mathbf{z}(t_i)\}, \{\mathbf{z}(\tau_j)\};\ \theta\right)$                 ▷ compute loss function

initialize $t = t_N$, $\mathbf{a} = \partial\mathcal{L}/\partial\mathbf{z}(t_N)$, $\mathbf{a}_\theta = 0$, $\mathbf{a}_t = \mathbf{a} \cdot f(\mathbf{z}(t_N), t_N;\ \theta)$, $\mathbf{z} = \mathbf{z}(t_N)$
**while** $t > t_0$ **do**
     $dt = \mathtt{AdpativeBackwardStepSize}(\mathbf{z}, \mathbf{a}, \mathbf{a}_\theta, \mathbf{a}_t, t, \theta)$      ▷ from ODE solver
     $(\tau_j, \mathbf{k}_j) = \mathtt{GetPreviousEvent}(\mathcal{H}, t)$        ▷ find previous event in sequence
     **if** $\tau_j < t - dt$ **then**
         $\mathbf{z}, \mathbf{a}, \mathbf{a}_\theta, \mathbf{a}_t = \mathtt{StepBackward}(\mathbf{z}, \mathbf{a}, \mathbf{a}_\theta, \mathbf{a}_t, dt, \theta)$     ▷ $1^{\mathrm{st}}$ term in Eq. (11), Eq. (9)
     **else**
         $dt = t - \tau_j$                           ▷ shrink step size
         $\mathbf{z}, \mathbf{a}, \mathbf{a}_\theta, \mathbf{a}_t = \mathtt{StepBackward}(\mathbf{z}, \mathbf{a}, \mathbf{a}_\theta, \mathbf{a}_t, dt, \theta)$
         $\mathbf{z}, \mathbf{a}, \mathbf{a}_\theta, \mathbf{a}_t = \mathtt{JumpBackward}(\mathbf{z}, \mathbf{a}, \mathbf{a}_\theta, \mathbf{a}_t, (\tau_j, \mathbf{k}_j), \theta)$    ▷ $2^{\mathrm{nd}}$ term in Eq. (11), Eq. (13)
     **end**
     $t = t + dt$
**end**

---