[Reviews · NeurIPS 2019]

Reviewer 1



I very much like this approach. It is well explained. Many of the existing approaches are overly complex cobblings of discrete-time and continuous-time ideas. By contrast, this is a clean solution that is computationally reasonable. The experimental results show a good variety of uses of the proposed method. It is here that the paper could be most improved: - The details of the MLPs (number of hidden units, number of layers, method for training) are completely omitted. - The paper does not compare with Neural ODE, particularly where the authors of that paper used an RNN to encode the starting latent value. - The paper does not do comparisons in Section 4.3 to other methods. The neural Hawkes process can be easily augmented (by adding another MLP from the hidden state) to predict marks (locations) associated with each event. Overall, I like the paper, but it really should supply more information on the experimental set-ups, in order to judge the significance of the effects shown. The "Reproducibility Response" claims this information is in the paper, but it is neither in the paper nor the supplementary material. One question which should be addressed in the paper: - Why are the RNN and Neural JSDE better than a Poisson process on Poisson data? If this is because of regularization, then this further underscores the need for clear experimental set-up descriptions so that this can be judged for all experiments and all methods.

Reviewer 2



This paper makes a notable extension to the Neural ODE work, by amending the continuous ODE with discontinuities introduced by discrete events. The technically difficulty is that in the presence of the discontinuity, the left and right limit of the adjoint vectors are different. The authors propose to handle the jumps of the adjoint vectors by lifting them from their right limit to the left. In terms of the model structure, both the dynamics function and the jump function are parameterized by MLP, which is also used to model the conditional intensity, and the emission. Various methods are compared empirically, showing the advantages of the proposed approach in accuracy, parsimonious of the model, and the capability of handling events with real-valued features. This paper is nicely written, well motivated, the relevant background and is presented clearly. I quite enjoyed reading the paper. Significance wise, I can see the proposed model having applications in a wide range of time series prediction tasks, such as financial, or retail demand time series where the spikes are triggered by certain events. Overall, a very good paper. Detailed comments and questions. 1. In section 3.1, it would be good to spell out the full generative model. For example, it would make it more clear how different marks are generated, and similarly with "events with real-valued features." 2. The original neural ODE can also be used to model inhomogeneous Poisson processes, I'm wondering how this method compares empirically to it? 3. For the experiments, it would be good to repeat multiple times and report the stds associated with the results. Also how's the running time, comparing to, for example, baseline RNN network, and RMTPP [20]?

Reviewer 3



This paper proposes Neural Jump Stochastic Differential Equations, a general framework for modeling temporal event sequences. We demonstrate the state of the art performance on prediction tasks. The paper is well written and model is interesting. My concern lies in the experiment section. The model does not show performance improvement on MIMIC dataset, and on stackoverflow the improvement is incremental, and it is questionable if it is statistical significant. Another evaluation metric is event time prediction, see the experiments in [20]. This evaluation metric will also demonstrate the predictive power of the proposed model. a relevant paper that should be cited: [1] A Stochastic Differential Equation Framework for Guiding Online User Activities in Closed Loop, Wang et al, AISTATS 2018 The derivations in section 3.1 should mention [1], where same equations have been derived. ---------------------------- thanks the authors for your response, and it addressed my concerns. I changed the score accordingly.

[Author Response · NeurIPS 2019]

We thank the reviewers for their careful reading of the paper. First, we would like to emphasize that all three reviewers agree the neural JSDE is an interesting model that introduces discrete events into continuous latent ODE framework. Reviewer 1 and Reviewer 2 further describe the model as clean/parsimonious, and they both state that neural JSDE have a very broad range of applications.

That being said, we would like to respond to the specific suggestions provided by Reviewers 1 and 2 on (1) including neural ODE as baselines and (2) details of the experimental setup; as well as the concerns raised by Reviewer 3 on (3) incremental prediction improvements on marker prediction and (4) the derivation of Eq. (11). The remaining minor points in the reviews should be easy to address in a final version of the paper.

**(1) Including the original neural ODE as a baseline.** Since neural ODE cannot model event effects, we thought comparing against it would be unfair. Essentially, neural ODE only captures Poissonian behavior in time series. That being said, we can still use the model to predict the conditional intensity of the point process datasets — we would just expect the model to only work well for Poisson process which does not depend on the event history. Indeed, the results in Table 1, which shows the mean absolute percentage errors (MAPE), demonstrates this. The accuracy of neural ODE for the Poisson process is on par with our neural JSDE. However, for the Hawkes process (Exponential), Hawkes process (Power-Law), and self-correcting process, neural ODE gives much larger predictions errors. Again, we should expect this behavior — a primary goal of neural JSDE is to account for the event history, a shortcoming of the original neural ODE framework.

Table 1: Neural ODE / JSDE predicted conditional intensity error.

| MAPE | ODE | JSDE |
|---|---|---|
| Poisson | 1.2 | 1.3 |
| Hawkes (E) | 172.0 | 5.9 |
| Hawkes (PL) | 91.4 | 17.1 |
| Self-Correcting | 27.2 | 9.3 |

**(2) Details on the experimental setup.** As we discuss here, neural JSDE introduces minimal additional architecture to the neural ODE framework. We will add the following details to the paper. For the point processes experiments, we used a 5-dimensional latent state and parameterized the dynamics function $f$, the jump function $w$, and the intensity function $\lambda$ using MLPs with one hidden layer and 20 hidden units. For the social/medical datasets, we used a 20/64-dimensional latent state and parameterized the functions with two-hidden-layer MLPs with 32/64 hidden units. The run time of neural JSDE is dominated by the underlying neural ODE dynamics and is therefore higher than the baseline RNN network, but the neural JSDE is much better suited for modeling complex dynamics and irregularly spaced time series, as demonstrated by its performance gain.

Reviewer 1 also noted the Poisson dataset does not fit well to the Poisson process. This is because the average sequence length in the Poisson dataset is relatively small (*e.g.* 20 events for Poisson *vs.* 200+ events for self-correcting process). The time series modeling software that we used is designed for long event sequences and ignores the idle time after the last event. We find that using longer Poisson sequences remedies this issue.

**(3) Results on marker prediction.** The main contribution of this work is introducing event handling into the continuous neural ODE framework with minimal computational overhead, while still maintaining its memory efficiency and ability to train end-to-end. We have demonstrated its strong performance in a range of settings in an attempt to highlight the flexibility of the framework. For the specific case of the Stack Overflow and MIMIC datasets, the baselines we compare against (RMTPP and neural Hawkes) are already quite strong, and prior advances in prediction on these datasets is also incremental [1, 2]. Our goal with this experiment is to demonstrate the modeling capability of neural JSDE rather than to blow competitive baselines out of the water. We will emphasize this in paper revisions.

**(4) Derivation of Eq. (11).** Reviewer 3 claims that Eq. (11) appears in [3]. We assume that Reviewer 3 is referring to the "SDE for Hawkes process" equation in section 3.2 of that paper. It turns out that Eq. (11) is considerably different. First, it uses a neural network to parameterize the continuous dynamics and jump; and second, it specifies the time evolution of latent state as opposed to directly modeling the conditional intensity or opinion. Moreover, the general idea of a "jump stochastic differential equation" is established with a long history in the financial mathematics literature. We felt that this was a standard-enough concept, but we are happy to include a reference to [3] in the final version of the paper, as it is certainly still relevant to our research.

# References

[1] Hongyuan Mei and Jason M Eisner. The neural hawkes process: A neurally self-modulating multivariate point process. In *Advances in Neural Information Processing Systems 30*. 2017.

[2] Nan Du et al. Recurrent marked temporal point processes: Embedding event history to vector. In *Proceedings of the 22nd ACM SIGKDD International Conference on Knowledge Discovery and Data Mining*, 2016.

[3] Yichen Wang et al. A stochastic differential equation framework for guiding online user activities in closed loop. In *Proceedings of the Twenty-First International Conference on Artificial Intelligence and Statistics*, 2018.


[Meta-Review · NeurIPS 2019]

This paper extends latent ODE-based time series models to include finite numbers of finite jumps, effectively incorporating Hawkes-process-like ideas into the framework. This breaks the determinism of latent ODE-based methods. Overall I feel that this work deserves to get in. However, I beseech the authors to change the name of the method to something that doesn't include the phrase "Stochastic Differential Equations". While this method does have differential equations as well as stochastic events, in my understanding SDEs already refer exclusively to processes driven by infinitesimal noise. Calling this model an SDE is a misleading and inaccurate. How about just "Neural ODEs with Jumps"?